# LncRNA LNC-565686 Promotes Proliferation of Prostate Cancer by Inhibiting Apoptosis through Stabilizing SND1

**DOI:** 10.3390/biomedicines11102627

**Published:** 2023-09-25

**Authors:** Xuke Qin, Jiacheng Zhong, Lei Wang, Zhiyuan Chen, Xiuheng Liu

**Affiliations:** Department of Urology, Renmin Hospital of Wuhan University, Wuhan 430060, China; drqinxk@163.com (X.Q.); drjaccy@whu.edu.cn (J.Z.); drwanglei@whu.edu.cn (L.W.)

**Keywords:** lncRNAs, SND1, prostate cancer, proliferation, apoptosis

## Abstract

Long non-coding RNAs (lncRNAs), typically more than 200 nt long, cannot encode proteins, but can regulate gene expression. They play an indispensable role in the occurrence and progression of various cancers. The main purpose of this study is to discuss the role and mechanism of LNC-565686 in prostate cancer. First, we found an increased expression of LNC-565686 in prostate cancer cells using RNA sequencing, which was further verified using qRT-PCR. Then, catRAPID was used to find that LNC-565686 might regulate SND1. Furthermore, a protein half-life experiment was performed to verify that LNC-565686 could stabilize the expression of SND1. In order to further explore the effects of LNC-565686 and SND1 on prostate cancer cells, we knocked down LNC-565686 and SND1 in prostate cancer cells, and verified using CCK8 and flow cytometry and western blot for the detection of apoptosis-related indicators. Collectively, we have found that LNC-565686 can promote the proliferation of prostate cancer cells and inhibit apoptosis by stabilizing the expression of SND1. Therefore, targeting LNC-565686 might be a new treatment for prostate cancer.

## 1. Introduction

Prostate cancer (PCa) is a leading cause of cancer-related deaths among men worldwide [1]. Thanks to prostate-specific antigen (PSA) testing, that can be used for diagnosis, the incidence of PCa is higher. Most early non-metastatic tumors can be treated with surgery and endocrinology, but treatment for advanced metastatic PCa is still limited [2,3]. Therefore, there is a need for a new therapeutic method that can improve patient stratification and treatment selection.

Long non-coding RNAs (lncRNAs) are RNA molecules that do not code for proteins but play important regulatory roles in gene expression, featuring no less than 200 nucleotides [4,5]. Their involvement in the development and progression of various cancers, including PCa, has been extensively studied [6].

Thanks to transcriptome sequencing, further research has been conducted on the early diagnosis [7] and grading of prostate cancer [8] and lncRNAs have emerged as promising candidates for biomarker discovery and therapeutic intervention in PCa [6]. Several studies have identified dysregulated lncRNAs in PCa tissues and cell lines, and their functional roles have been investigated in vitro and in vivo [9]. Some lncRNAs have been shown to promote PCa progression by regulating cell proliferation, invasion, migration, and metastasis, while others act as tumor suppressors by inhibiting these processes [10,11]. Moreover, SND1 have been implicated in the resistance to androgen deprivation therapy, which is the standard treatment for advanced PCa [12]. Therefore, understanding the role of lncRNAs in PCa pathogenesis may pave the way for the development of new diagnostic and therapeutic strategies.

Staphylococcal nuclease domain-containing 1 (SND1), also known as “Tudor staphylococcal nuclease” or “P100”, a multifunctional protein, plays a multifaceted role as a transcriptional coactivator or is involved in the processing of precursor messenger RNA, whose expression is widely conserved in humans and other species. Recently, it was reported that SND1 plays essential roles in PCa progression and drug resistance [13,14].

Here, in the established LncRNAs RNA-sequencing of tumor tissues and adjacent normal tissues, we distinguished LNC-565686 (Ensemble ID: ENST00000565686, also termed as RP11-24M17.4 or LOC101929408) as a lncRNA that was significantly up-regulated in PCa. We also revealed that LNC-565686 is highly expressed in PCa cells, promoting PCa development by inhibiting tumor cell apoptosis. Meanwhile, we demonstrated that LNC-565686 further affects the progression of PCa by regulating the stability of the SND1 protein. The above results are expected to provide new ideas for revealing the new mechanism of lncRNAs in PCa and potential targets for the clinical diagnosis and treatment of PCa. 

## 2. Materials and Methods

### 2.1. Patients and Specimens

A total of 12 pairs of prostate cancer tissues and adjacent normal tissues were obtained from patients who underwent urological surgery in Renmin Hospital of Wuhan University from 2020 to 2021. All research protocols were approved by the Ethics Committee of Renmin hospital of Wuhan University (ethic code: WDRY2019-K084). Informed consent was obtained from patients before surgery and the collection of relevant clinical data. All tissue samples were promptly frozen in liquid nitrogen and then stored at −80 °C until analysis.

### 2.2. RNA Sequence Analysis

Three pairs of prostate cancer and adjacent normal tissues were used for RNA sequencing. The samples were processed and sequenced by Sangon Biotech (Wuhan, China). The results were processed using the R software (version 4.1.3) with limma package. *p* values were adjusted using Benjamini–Hochberg. The differentially expressed LncRNAs were defined as |logFC| > 2 and *p* adjust <0.05.

### 2.3. Cell Culture and Transfection

RWPE-1, LNCaP, and PC3 cells were purchased from the ATCC (American Type Culture Collection). Cells were cultured in RPMI 1640 medium supplemented with 10% fetal bovine serum (GIBCO) at 37 °C with 5% CO_2_. The process of constructing plasmids and transfecting cells involved extracting plasmids using a Small Plasmid Extraction Kit (Transgen BioTech, Beijing, China) after single colonies were vaccinated and amplified, and the bacterial fluid was verified through sequencing. To obtain lentiviral containing the target gene, 293 T cells were co-transfected with pLVX-SND1-ZsGreen-Puro (recombinant plasmid) using a Lentiviral packaging kit (Viraltherapy Technologies, Wuhan, China). Transfection was carried out by transferring 5 × 10^5^ cells/mL cells to cell plates with an MOI of 20. After two days, lentivirus-infected cells were selected using complete medium containing 10 μg/mL of puromycin. For siRNA, it was constructed and obtained from Sangon Biotech (Wuhan, China) and transfected into cells using Lipo8000 (Beyotime, Shanghai, China) for at least 48 h before harvesting for RNA and protein preparation. The sequences of SiRNAs are listed in Table 1.

### 2.4. qRT-PCR

Total RNA was extracted from cell cultures using a FastPure Cell/Tissue Total RNA Isolation Kit (Vazyme, Nanjing, China) according to the manufacturer’s instructions. RNA concentration was measured using a NanoDrop spectrophotometer (Thermo Scientific, Shanghai, China). cDNA was synthesized from RNA using the HiScript III All-in-one cDNA Synthesis Kit (Vazyme, Nanjing, China). qPCR was performed using TB Green Premix Ex Taq (Takara, Beijing, China) on a recommended Real-Time PCR System. The primer sequences used for qPCR are listed in Table 2. The cycling conditions were as follows: initial denaturation at 95 °C for 30 s, followed by 40 cycles of denaturation at 95 °C for 5 s and annealing/extension at 60 °C for 20 s. The expression levels of the target genes were normalized to the expression levels of the housekeeping gene β-actin using the ΔΔCt method.

### 2.5. Bioinformatics Analysis

The online database GEPIA (http://gepia.cancer-pku.cn/index.html (accessed on 20 September 2022)) was used to analyze the different mRNA expression levels of SND1 in PC tissues and normal tissues. The online tools catRAPID (http://s.tartaglialab.com/catrapid/omics (accessed on 17 September 2022)) was used to predict the binding proteins of LNC-565686. Gene Set Enrichment Analysis (GSEA) was performed to identify differentially enriched gene sets in our experimental group compared to the control group. The gene expression data were obtained from the ICGC database (PRAD-CA). The analysis was performed using the GSEA software (version 4.1.0) obtained from the Broad Institute (http://www.broadinstitute.org/gsea (accessed on 13 September 2022)). The Molecular Signatures Database (MSigDB) was used as the reference gene set.

### 2.6. Western Blot Analysis

Protein was extracted from cells using RIPA buffer (Beyotime, Shanghai, China) containing PMSF (Beyotime, Shanghai, China). Protein concentration was determined using a BCA Protein Assay Kit (Thermo Fisher Scientific, Shanghai, China). Equal amounts of protein were separated by SDS-PAGE and transferred to a polyvinylidene difluoride membrane (Millipore, Burlington, MA, USA). The membrane was blocked with 5% non-fat dry milk in Tris-buffered saline with 0.1% Tween 20 (TBST) for 1 h at room temperature and incubated with the primary antibody overnight at 4 °C. Primary antibodies used here were polyclonal rabbit antibodies against SND1 (1:1000 dilution; Affinity, Liyang, China), Bcl-2 (1:1000 dilution; Affinity, Liyang, China), BAX (1:1000 dilution; Affinity, Liyang, China), C-Caspase-3 (1:500 dilution; Affinity, Liyang, China), and β-actin (1:1000 dilution; Affinity, Liyang, China). The membrane was then washed with TBST and incubated with the appropriate secondary antibody for 1 h at room temperature. The protein bands were visualized using enhanced chemiluminescence reagents (Thermo Fisher Scientific, Shanghai, China) and quantified using ImageJ software (version 1.52).

### 2.7. Cell Viability

Cell viability was evaluated using a CCK-8 assay (Beyotime, Shanghai, China) following the manufacturer’s instructions. Briefly, cells were seeded into 96-well plates at a density of 5 × 10^3^ cells/well and incubated overnight. Subsequently, the cells were transfected with SiRNA or plasmid vector. Following this, the cells were incubated with CCK-8 reagent and cell viability was measured through absorbance readings using a microplate reader (Molecular Devices, Sunnyvale, CA, USA) at 450 nm.

### 2.8. Apoptosis Analysis

Cells were harvested and stained using a Annexin V-FITC/PI Apoptosis Detection Kit (Vazyme, Nanjing, China) according to the manufacturer’s protocol. The percentage of FITC-positive apoptotic cells was analyzed using CytoFLEX S (Beckman Coulter, Brea, CA, USA).

### 2.9. Statistical Analysis

All data were expressed as mean ± SD and analyzed using SPSS software (version 25.0). Statistical significance was determined using unpaired two-tailed Student’s *t*-tests or one-way analysis of variance (ANOVA) followed by Tukey’s multiple comparisons test. *p*-values less than 0.05 were considered statistically significant.

## 3. Results

### 3.1. The RNA Sequencing and Identification of Differently Expressed lncRNAs

In order to identify the expression of LncRNAs in PCa, we collected 12 cases of PCa and its adjacent normal tissues, and selected three pairs of them for RNA sequencing. The results showed that 441 LncRNAs were abnormally expressed (|LogFC| > 2, *p* value < 0.05), including 156 up-regulated LncRNAs and 285 down-regulated LncRNAs (Figure 1A). In order to further verify the results, our research group selected the top 10 LncRNAs with the most significant up-regulation and down-regulation for confirmation in tissues and cell lines. The result of Real-time PCR detection of LNC-565686 showed that the expression of LNC-565686 was significantly increased in the cancer tissues and two PCa cells (Figure 1B). LNC-565686 is a transcript that is located in the exon region of gene ENSG00000260288 (Figure 1C). Furthermore, LNC-565686 was confirmed to be mainly located in the nucleus by nucleo-plasmic separation experiments (Figure 1D).

### 3.2. The Inhibition of Apoptosis of PCa Cells by LNC-565686

First of all, the gene set enrichment analysis (GSEA) of prostate cancer data from ICGC showed that the apoptosis pathway in the LNC-565686 down-regulated group was significantly enriched (Figure 2A). Therefore, we speculated that LNC-565686 may inhibit the apoptosis of PCa cells. Given the knockdown of LNC-565686 in PCa cell lines, loss-of-function studies were performed in LNCaP cells and PC3 cells. A CCK-8 cell-counting assay was performed after cells were transfected with siRNA targeting LNC-565686 (SiLNC-565686) or negative control SiRNA (SiNC). The results indicated that the knockdown of LNC-565686 reduced the cells’ proliferative activity in vitro (Figure 2B,C). In addition, flow cytometry was performed to detect the effect of LNC-565686 on the apoptosis of PCa cells. As shown in Figure 2D, the proportion of apoptotic cells in the SiLNC-565686 groups was higher compared with the SiNC groups, which verified that the apoptosis of PCa cells was notably promoted by LNC-565686 inhibition (Figure 2D). Moreover, Western blotting analysis also showed that apoptosis-related proteins, such as Bax/Bcl-2, were upregulated in the SiLNC-565686 group (Figure 2E). In summary, these results showed that LNC-565686 promotes PCa progression by regulating cell proliferation and apoptosis.

### 3.3. Prediction and Screening of LNC-565686 Target Protein

Firstly, we used catRAPID, an online prediction application, to predict the proteins interacting with LNC-565686. From these, we selected the top ten proteins with the highest scores, as shown in Figure 3A. Among them, SND1 aroused our interest due to its high expression in multiple tumor tissues [15]. The binding of SND1 and LNC-565686 was predicted in catRAPID (Figure 3B). In addition, we found that SND1 was also up-regulated in PCa tissues through the GEPIA database (Figure 3C). And, according to previous reports, Circular RNA METTL9 can bind to SND1 and upregulate its expression [16]. So we speculated that LNC-565686, also a non-coding RNA, may also be able to interact with SND1. To investigate whether there is a regulatory relationship between LNC-565686 and SND1, we detected the mRNA and protein levels of SND1 after LNC-565686 knockdown. The results showed that SND1 protein was down-regulated in PC3 cells and LNCaP cells with LNC-565686 knockdown (Figure 3D), while the level of SND1 mRNA had no changes (Figure 3E). Furthermore, by performing Cycloheximide (CHX), a protein synthesis inhibitor for protein half-life experiment, we demonstrated that, in LNC-565686 knockdown cells, the half-life of SND1 protein decreased, indicating that its stability decreased (Figure 3F). At the same time, it was further verified that LNC-565686 had a certain positive effect on promoting the stable expression of the SND1 protein.

### 3.4. Inhibition of Apoptosis of PCa Cells by SND1

First of all, we used CCK-8 cell-counting assay after cells were transfected with siRNA targeting SND1 (SiSND1). The results suggested that the knockdown of SND1 down-regulated the cells proliferative activity in vitro (Figure 4A). In addition, flow cytometry was performed to detect the effect of SND1 on the apoptosis of PCa cells. As shown in Figure 4B, the proportion of apoptotic cells in the SiSND1 groups was higher in comparison to the SiNC groups. The results further verified that the apoptosis of PCa cells was prominently accelerated by SND1 suppression (Figure 4B). Moreover, western blotting analysis also showed that apoptosis-related proteins, such as Bax/Bcl-2 and Cleaved caspase-3, were upregulated in the SiSND1 group (Figure 4C,D). To sum up, these results certified that SND1 promotes PCa progression by regulating cell proliferation and apoptosis.

### 3.5. Influence on Biological Function of PCa Cells by LNC-565686 Regulating of SND1

To further demonstrate that the up-regulation of SND1 is the mediator of LNC-565686 in PCa cells, rescue experiments were performed by co-transfecting LNC-565686 siRNA and SND1 vectors into PCa cells. The transfection efficiency was measured using qRT-PCR and western blot (Figure 5A). The results of CCK-8 and flow cytometry showed that both the attenuated proliferation and enhanced apoptosis of PCa cells induced by LNC-565686 knockdown could be rescued by the overexpression of SND1 (Figure 5B,C). These data suggested that SND1 is a functional target of LNC-565686 in PCa.

## 4. Discussion

As there has been extensive progress in the diagnosis and treatment of PCa in recent decades, patients have achieved fantastic curative effects through ADT (androgen deprivation therapy) [3]. Nevertheless, the recurrence and drug resistance of PCa are still the thorny problems affecting the prognosis of patients [17]. Hence, it is pressing to discover the detailed molecular mechanism of PCa, and thus find a way to boost the benefits of therapy. In recent years, more and more evidence has shown that lncRNAs are closely related to the development and drug resistance of PCa. Ruchi Ghildiyal et al. found that the loss of lncRNA NXTAR in PCa improves androgen receptor expression and enzalutamide resistance [12], while Simeng Wen et al. discovered that lncRNA NEAT1 promotes bone metastasis of PCa through N6-methyladenosine [10].

Long noncoding RNAs (lncRNAs) are non-coding RNAs with a length of over 200 nt and no obvious open reading frames (ORFs) [18,19]. With continuous in-depth research, lncRNAs have been found to play an indispensable role in cell proliferation, drug resistance, metabolism, apoptosis, senescence, and other myriad biological processes [20,21,22,23]. LncRNAs participate in these processes by coordinating enzymatic protein to regulate or degrade or by fine-tuning different chromatin states, which also indicates that the disorder of lncRNAs will cause the occurrence and development of many diseases. Currently, some lncRNAs have been found to promote the development of cancer in PCa, while others have inhibitory effects. For example, long non-coding RNA NEAT1 was verified to promote bone metastasis of PCa through N6-methyladenosine [10], whereas LncRNA MEG3 was found to inhibit the progression of PCa by the regulating miR-9-5p/QKI-5 axis [10].

The results of our study focused on detecting the effect of LNC-565686 in PCa; we revealed that LNC-565686 was an up-regulated lncRNA in PCa through RNA high-throughput sequencing. The lncRNA has not been studied thus far and we are the first to explore the role of LNC-565686 in PCa. With the help of many biological function experiments, we have verified that LNC-565686, as an oncogene, affects the progression of PCa.

In order to further verify whether LNC-565686 is consistent with the characteristics of lncRNAs, we discovered that lncRNA-565686 has the characteristics of genes larger than 200 nt and cannot encode proteins through ORF Finder (https://www.ncbi.nlm.nih.gov/orffinder/ (accessed on 1 September 2022)) analysis. Not only did LNC-565686 conform to the basic characteristics of long non-coding RNAs, but nucleo-plasmic separation experiments were used to show that LNC-565686 was mainly located in the nucleus.

Further, we explored the mechanism of LNC65686 in PCa, and found that LNC-565686 was related to tumor apoptosis through GSEA. At the same time, proliferation and apoptosis experiments were performed by comparing the knockdown LNC-565686 and the control group in the PCa cells. The results showed that the proliferation of PCa cells was significantly weakened while apoptosis was enhanced after LNC-565686 knockdown. Conversely, this also suggests that LNC-565686 promotes the progression of PCa by regulating cell proliferation and apoptosis.

In addition, using the prediction of catRAPID, we selected the top ten likely interacting proteins.

SND1 has been shown to promote tumor development in many tumors. For example, the carcinogenic effects of SND1 in human tumors were identified in a pan-cancer analysis, which showed that SND1 was found to promote cell proliferation and tumor progression through inducing mitochondrial autophagy [15]. In addition, SND1 has been demonstrated to promote the metastasis of clear cell renal cell carcinoma by mediating ERK signaling and epithelial–mesenchymal transformation [24]. It has also been found in PCa that SND1 interacts with Circ_0004087 and promotes docetaxel resistance in PCa by promoting the mitotic error correction mechanism [13].

Based on the above studies, we speculated that LNC-565686 could affect the proliferation and apoptosis of PCa by acting on SND1 and verified the effect of SND1 knockdown on the proliferation and apoptosis of PCa cells. Further, we conducted protein half-life experiments by means of the protein synthesis inhibitor Cycloheximide (CHX), and found that the knockdown of LNC-565686 resulted in the decreased stability of the SND1 protein. In addition, LNC-565686 siRNA and SND1 vector were co-transfected into PCa cells for a rescue experiment, which verified that LNC-565686 siRNA promotes the occurrence and development of PCa by stabilizing the expression of SND1.

At present, ADT treatment is still the first line treatment for advanced PCa. However, after a certain period of ADT treatment, cancer cells usually develop drug resistance to ADT. Therefore, it is urgent to find a new therapeutic target for castration-resistant PCa. Currently, lncRNAs have received increasing attention and in-depth research, and lncRNAs are also considered to be closely related to ADT [25] and chemotherapy resistance [26] in PCa. In our study, we found that LNC-565686 expression was abnormally elevated in PCa and had a promoting effect on the viabilities of PCa cells. Inhibiting LNC-565686 expression in PCa contributes to inhibiting the proliferation of PCa cells. It may provide a new target for the treatment of advanced PCa. We believe that LNC-565686 is a potential and promising therapeutic target for clinical PCar patients. However, there are still limitations at present. In our study, although LNC-565686 expression was suppressed, PCa cells remained proliferative. Therefore, its application still needs to be verified in subsequent experiments, such as determining whether to inhibit LNC-565686 alone or in combination with ADT, docetaxel, etc.

## 5. Conclusions

In summary, our study revealed, for the first time, that LNC-565686 is an up-regulated lncRNA in human PCa, which can regulate the expression of SND1. We also demonstrated that the knockdown of LNC-565686 can effectively inhibit the proliferation and promote the apoptosis of PCa cells by reducing the stability of the SND1 protein. These findings suggest that LNC-565686 can serve as a brand new therapeutic target for PCa in the future.

## Figures and Tables

**Figure 1 biomedicines-11-02627-f001:**
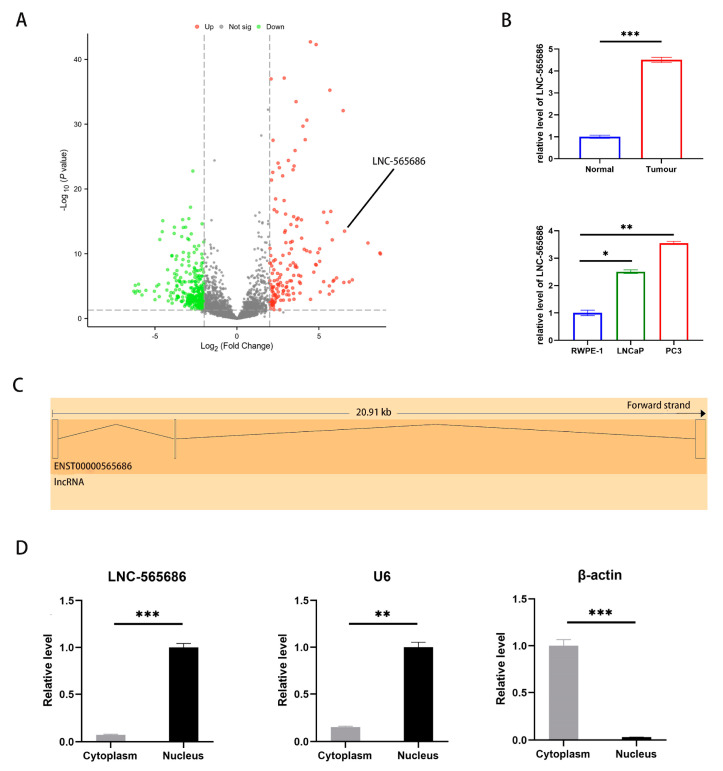
LNC-565686 is overexpressed in prostate cancer cells (**A**) RNA sequencing was used to verify the expression of LncRNAs in PCa; (**B**) Real-time PCR was performed to further verify the expression of LNC-565686 in both prostate cancer tissues and cells; (**C**) The figure of transcript of ENST000000565686 from ensemble database; (**D**) Nucleo-plasmic separation experiments were performed to locate the position of LNC-565686 in prostate cancer cells. (Data are expressed as mean ± SD of at least three experiments. *, *p* < 0.05, **, *p* < 0.01, ***, *p* < 0.001).

**Figure 2 biomedicines-11-02627-f002:**
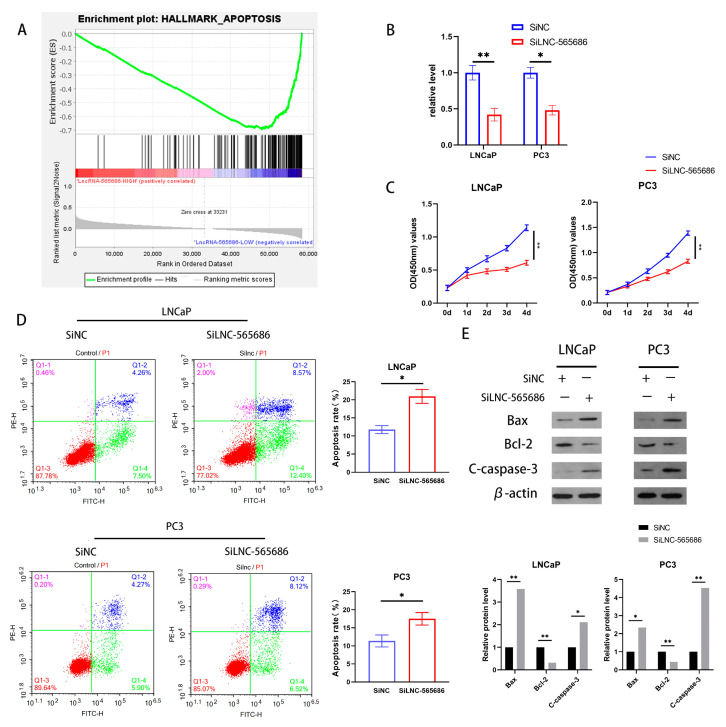
LNC-565686 down-regulation inhibits apoptosis of PCa (**A**) Cell apoptosis was enriched in the LNC-565686 high expression group of PCa through GSEA; (**B**) The mRNA levels of LNC-565686 was reduced in LNC-565686 knockdown PCa cells through qRT-PCR assays; (**C**) LNC-565686 knockdown restrained PCa cell proliferation through CCK8 assay; (**D**) LNC-565686 knockdown increased PCa cells apoptosis rate via cell flow cytometry; (**E**) Western blotting analysis and corresponding quantitative analysis were performed to explored the influence of LNC-565686 on the expression of apoptosis markers. (Data are expressed as mean ± SD of at least three experiments. * *p* < 0.05, **, *p* < 0.01).

**Figure 3 biomedicines-11-02627-f003:**
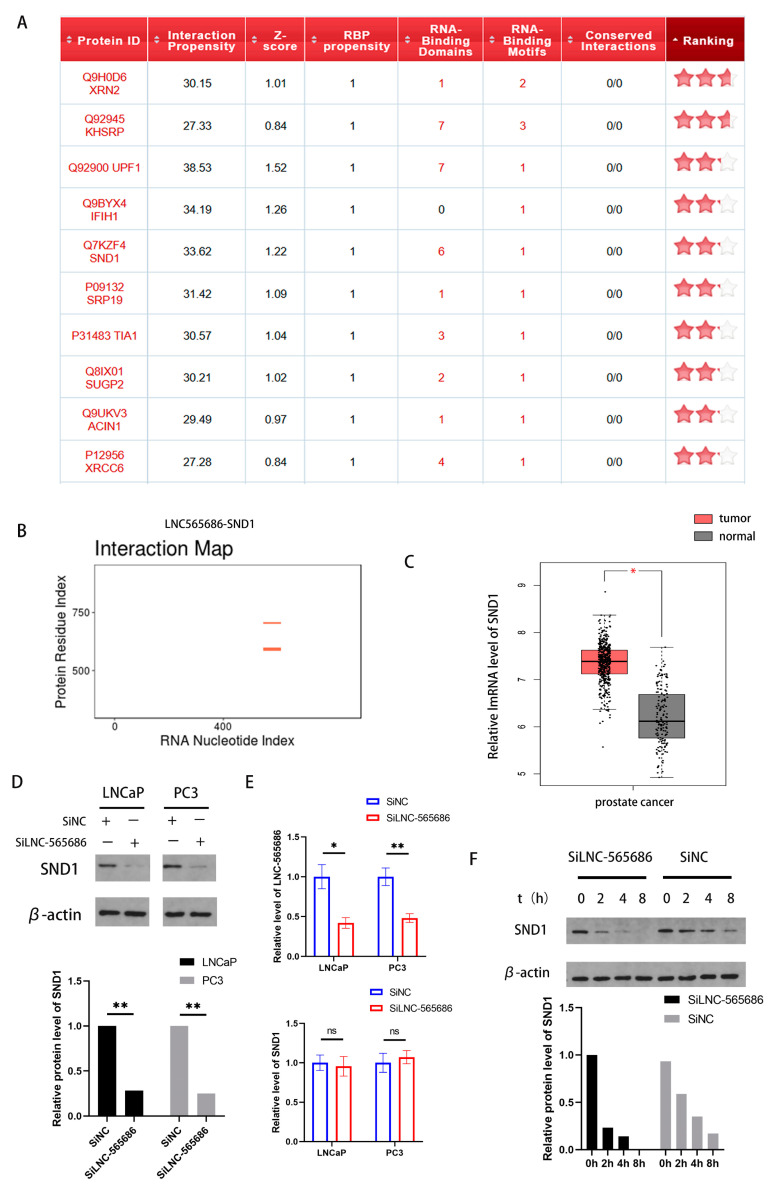
LNC-565686 regulates the expression and stability of SND1 (**A**) CatRAPID was used to select the top 10 proteins with the highest scores interacting with LNC-565686; (**B**) The binding of SND1 and LNC-565686 was predicted via catRAPID; (**C**) SND1 expression level was up-regulated in PCa tissues compared to normal tissues according to the TCGA database; (**D**,**E**) Western blotting analysis was performed to show the knockdown efficiency of SND1 in protein (**D**) or RNA (**E**) levels in PCa cells transfected with SiNC and SiLNC-565686; (**F**) Protein half-life experiment was used to verify that the SND1 protein half-life of 0 to 8 h in PCa cells transfected with SiNC and SiLNC-565686. (Data are expressed as mean ± SD of at least three experiments. *, *p* < 0.05, **, *p* < 0.01, ns, no significance).

**Figure 4 biomedicines-11-02627-f004:**
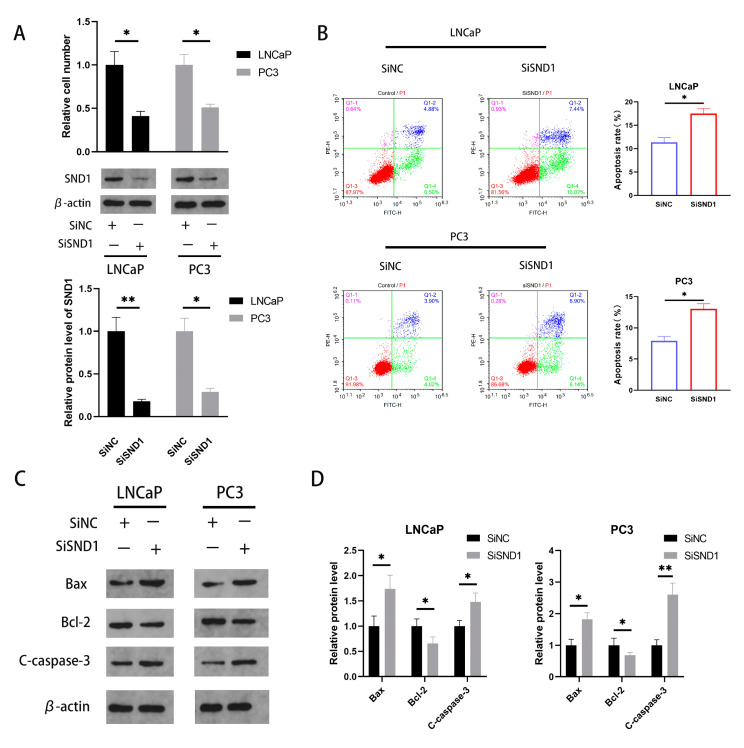
SND1 down-regulation inhibits apoptosis of PCa. (**A**) The protein level of SND1 was reduced in SND1 knockdown PCa cells through western blotting analysis; (**B**) SND1 knockdown increased PCa cells apoptosis rate via cell flow cytometry; (**C**) Western blot was performed to explore the influence of LNC-565686 on the expression of apoptosis markers; (**D**) Quantitative analysis was performed to display the protein expression levels of apoptosis markers in PCa cells. (Data are expressed as mean ± SD of at least three experiments. * *p* < 0.05, **, *p* < 0.01).

**Figure 5 biomedicines-11-02627-f005:**
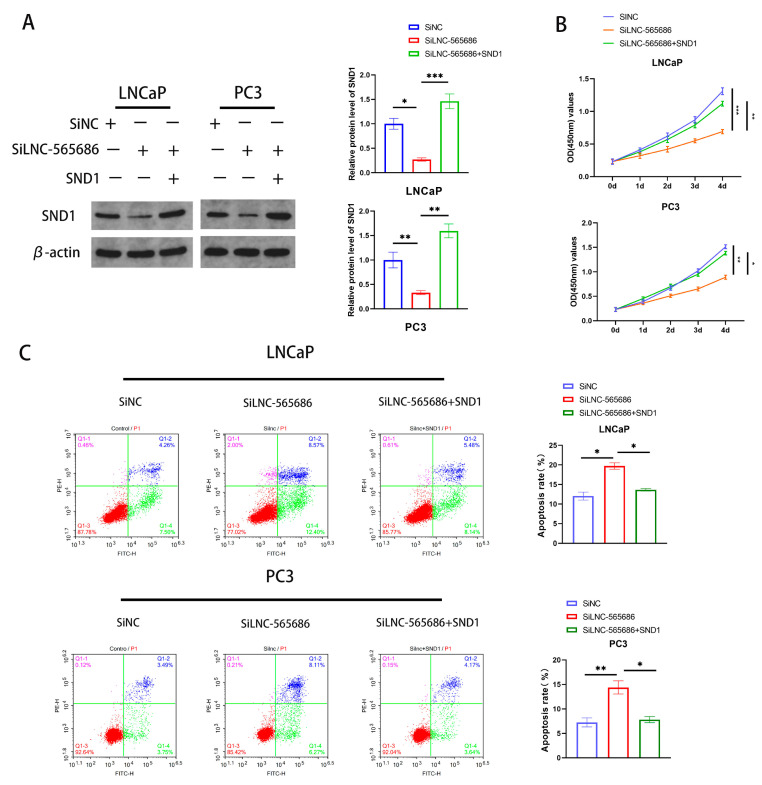
LNC-565686 influences on biological function of PCa cells by regulating of SND1. (**A**) Western blotting analysis was performed to show the overexpression efficiency of SND1 in protein levels in PCa cells co-transfected with LNC-565686 siRNA and SND1 vector; (**B**) CCK8 assay was used to measure the proliferation levels of PCa treated with SiNC, SiLNC-565686, and SiLNC-565686 + SND1; (**C**) Overexpression of SND1 saved the increased apoptosis of PCa cells caused by LNC-565686 knockdown via cell flow cytometry. (Data are expressed as mean ± SD of at least three experiments. *, *p* < 0.05, **, *p* < 0.01, ***, *p* < 0.001).

**Table 1 biomedicines-11-02627-t001:** Sequences of SiRNA.

Name	Sense	Antisense
SiLNC-565686	UGCAUAAAGUUGAGGAACATT	UGUUCCUCAACUUUAUGCATT
SiSND1	GCAACAUUCGAGCUGGAAATT	UUUCCAGCUCGAAUGUUGCTT
SiNC	UUCUCCGAACGUGUCACGUTT	ACGUGACACGUUCGGAGAATT

**Table 2 biomedicines-11-02627-t002:** Primer sequences used for qPCR.

Target Gene	Forward Primer	Reverse Primer
**LNC-565686**	AAATCCACACACCCAGAACATCTCG	TGGCGTCTCCTCCTATGTCTTCC
**SND1**	CAGAACCGGCTTTCAGAATGT	TAGTATGTGAACCGTTCCCCT
**β-actin**	CATGTACGTTGCTATCCAGGC	CTCCTTAATGTCACGCACGAT
**U6**	CTCGCTTCGGCAGCACA	AACGCTTCACGAATTTGCGT

## Data Availability

The data presented in this study are available on request from the corresponding author.

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
