# Peer review of "LncRNA LNC-565686 Promotes Proliferation of Prostate Cancer by Inhibiting Apoptosis through Stabilizing SND1"

_biomedicines, 2023, doi:10.3390/biomedicines11102627_

Round 1

Reviewer 1 Report

Qin et al started by finding a novel lncRNA LNC-565686 which is overexpressed in prostate cancer, followed by investigating its regulation of SND1, a protein that plays essential roles in PCa progression and drug resistance. By knocking down LNC-565686 or SND1, the cancer cells become more apoptotic and their growth is slowing down, but the cells are still proliferative, which may need some explanation for the effectiveness of LNC-565686 as a therapeutic target. Some minor issues: (1)full name of siNC when it first appeared in the paper; sequences or sources for siNC, siLNC-565686 and siSND1; (2)number of biological replicate, error bar and statistical representation of "*" etc. in Figure 1B and 1D; (3)Figure 3E and 3D switched at line 189-190; (4)missing Y-axis in Figure 3C, which could be "SND1" according to texts at line 183 (which is more likely), or "LNC-565686" according to Figure 3C legend; (5)Need protein control such as beta-actin for Figure 4A; (6)a typo "lncRNA-56568" at line 264.

Reviewer 2 Report

The study investigates LNC-565686 in prostate cancer, finding higher expression in cancer cells using RNA sequencing and qRT-PCR. The lncRNA interacts with SND1, influencing its stability as confirmed by protein half-life tests. The results indicate that LNC-565686 promotes prostate cancer cell growth and inhibits apoptosis by stabilizing SND1 expression, (specially knowing-down). Targeting LNC-565686 might offer a new approach to diagnosing/treating prostate cancer and understanding the progression of the disease.

The manuscript is well-presented, and the methods are adequately applied. However, I have some minor suggestions/concerns:

- Earlier in the introduction, the authors may refer to general papers that try to find diagnostics/prognosis biomarkers for Pca, I suggest highlighting PMID: 30890858, PMID: 31835700, or similar studies.

- In the "RNA sequence analysis", I wish the authors may check the FDR using Benjamin-Hochberg or similar techniques and validate whether the P-value was correctly calculated.

- In the discussion, the authors may highlight the importance of the results from the clinical practice point of view.

- please fix the following:

Funding: This research was funded by NAME OF FUNDER, grant number XXX, and the APC was 301 funded by

Reviewer 3 Report

"LncRNA LNC-565686 promotes proliferation of prostate cancer 1 by inhibiting apoptosis through stabilizing SND1" is quite interesting study, however I have several concerns:

1. Figure 4A and Figure 5C. Westerns do not show significant difference and have to be repeated.

2. Figure 3 A & B. Figures should be at least visible for the reader.  Even as a reviewer I did not have enough time tu use binocular in order to decipher results.

3.  Do the colors in Figures 2D, 4B and 5C have to be so annoying? 

jus have a look once more
